# Novel QTL for Low Seed Cadmium Accumulation in Soybean

**DOI:** 10.3390/plants11091146

**Published:** 2022-04-24

**Authors:** Nour Nissan, Julia Hooker, Arezo Pattang, Martin Charette, Malcolm Morrison, Kangfu Yu, Anfu Hou, Ashkan Golshani, Stephen J. Molnar, Elroy R. Cober, Bahram Samanfar

**Affiliations:** 1Agriculture and Agri-Food Canada, Ottawa Research and Development Centre, Ottawa, ON K1A 0C6, Canada; nournissan96@gmail.com (N.N.); julia.hooker@agr.gc.ca (J.H.); arezo.p@hotmail.ca (A.P.); martin.charette@agr.gc.ca (M.C.); malcolm.morrison@agr.gc.ca (M.M.); steve.molnar@agr.gc.ca (S.J.M.); elroy.cober@agr.gc.ca (E.R.C.); 2Department of Biology, Ottawa Institute of Systems Biology, Carleton University, Ottawa, ON K1S 5B6, Canada; ashkan_golshani@carleton.ca; 3Agriculture and Agri-Food Canada, Harrow Research and Development Centre, Harrow, ON N0R 1G0, Canada; kangfu.yu@agr.gc.ca; 4Agriculture and Agri-Food Canada, Morden Research and Development Centre, Morden, MB R6M 1Y5, Canada; anfu.hou@agr.gc.ca

**Keywords:** soybean (*Glycine max*), QTL analysis, *Cda1*, markers, seed cadmium (Cd), CAP/dCAP markers, KASP

## Abstract

Soybean is a valuable crop, used in animal feed and for human consumption. Selecting soybean cultivars with low seed cadmium (Cd) concentration is important for the purpose of minimizing the transfer of Cd into the human body. To ensure international trade, farmers need to produce soybean that meets the European Union (EU) Cd limit of 0.2 mg kg^−1^. In this study, we evaluated two populations of recombinant inbred lines (RILs), X5154 and X4050, for seed Cd accumulation. Linkage maps were constructed with 325 and 280 polymorphic simple sequence repeat (SSR) markers, respectively, and used to identify a novel minor quantitative trait locus (QTL) on chromosome 13 in the X4050 population between SSR markers Satt522 and Satt218. Based on a gene ontology search within the QTL region, seven genes were identified as candidates responsible for low seed Cd accumulation, including *Glyma.13G308700* and *Glyma.13G309100*. In addition, we confirmed the known major gene, *Cda1*, in the X5154 population and developed KASP and CAPS/dCAPS allele-specific markers for efficient marker-assisted breeding for *Cda1*.

## 1. Introduction

Cadmium (Cd) is a toxic metal in our environment with harmful effects on humans, as well as deleterious effects on plant growth and development [1]. High Cd concentrations in the soil, due to anthropogenic as well as natural sources, can change plant growth at both the morphological and physiological levels [2]. Some of these effects include delayed growth rate, increased oxidative damage, leaf chlorosis, inhibition of respiration, stunted growth, and decreased ability in nutrient uptake [3,4,5,6]. More importantly, Cd has been shown to negatively impact the human physiology, including harming the male and female reproductive systems by weakening spermatogenesis, sperm quality and motility, damaging reproductive hormone balance, as well as negatively affecting menstrual cycles [7]. Cd binds to mammalian tissue, particularly the liver and kidney, causing organ failure and cancer [8]. Studies in animals have shown that Cd and its compounds can trigger benign and malignant tumor formation, *itai-itai* disease, as well as pancreatic cancer [8,9,10].

Soybean (*Glycine max* (L.) Merr.) is an essential part of the global diet. Soybean can benefit a healthy diet; however, a limitation to the supply of high-quality soybean is the accumulation of Cd in seeds. To ensure health and continued stability of the export markets, farmers need to overcome the challenge of high Cd concentration in soybean seeds. High concentration of Cd in crops is not only an issue in soybean; it has also been recorded in staple food crops, such as durum wheat (*Triticum durum* var. *durum* Desf.), potato (*Solanum tuberosum* L.), and rice (*Oryza sativa* L.) [11]. During the 1990s, Canadian durum wheat exports were impaired due to cultivars having a seed Cd uptake greater than the European Union (EU) limit [12]. This led to extensive research resulting in the identification of a single dominant gene that controlled low Cd accumulation in durum wheat seed [11]. To ensure that soybean farmers are not subjected to the same fate, new cultivars need to be developed that will meet the trade regulations established by the EU and UNICEF, currently set at a maximum of 0.2 mg kg^−1^ of Cd in soybean [13,14].

The application of contaminated fertilizers (such as rock phosphate) to fields, in addition to manure and sewage sludge (which are products of animals consuming and accumulating Cd), can increase Cd concentration in soils where previously low Cd concentrations existed [15]. In environments where it is more bioavailable, Cd uptake by plants increases. A mean seed Cd level of 0.15 (±0.044) mg kg^−1^ was found across 30 soybean varieties grown in southern China in unpolluted soil with a Cd level of 0.15 mg kg^−1^ [16,17]. However, in northeastern China, a 4-year-long study determined mean Cd seed concentrations as low as 0.033 mg kg^−1^ across 511 samples of different cultivars [17,18]. Soil Cd level is an important consideration for seed Cd accumulation; low soil pH and high soil contamination contribute to increased Cd in soybeans [17]. Countries including China and France have reported soil Cd levels exceeding 100 mg kg^−1^ [19,20,21]. In a large-scale agricultural survey from 2011 to 2013, the Japan Ministry of Agriculture, Forestry and Fisheries found over 49.5% of soybean seed samples (*n* = 1800) had over 0.1 mg kg^−1^ Cd, and 9.6% of samples exceeded the maximum limit of 0.2 mg kg^−1^ Cd [17].

Soybean cultivars can be classified as either high or low Cd accumulators. The amount of Cd taken up by a plant depends on the concentration of the available Cd in the soil, as demonstrated by Morrison et al. [22] who identified a strong correlation between available soil Cd concentration and seed concentration. Of the Canadian food-grade soybean cultivars that were tested, approximately 70% were high Cd accumulators, and 30% were low Cd accumulators. Further research led to the development of a controlled-environment screening method that can be used to accurately select low Cd-accumulating cultivars [23]. A major QTL for seed Cd content was reported independently by two different research groups in different germplasms [24,25]. This QTL explained 57% of the phenotypic variation in the population studied by Ref [24] and 82, 57, and 75% of the parental variation in the F_6:7_, F_7:8_, and F_8:9_ generations of the population, respectively, studied by Ref [25]. The name *Cda1* has been proposed for the locus, and the underlying gene *Glyma09g06170* was identified by Jegadeesan et al. [24]. However, no allele-specific markers have been developed for it to date at the gene level.

Our objectives were to identify a QTL for seed Cd accumulation in soybean RIL populations, to look for candidate genes underlying the novel QTL, and to develop allele-specific markers for efficient marker-assisted breeding of *Cda1*.

## 2. Results

### 2.1. Population Phenotyping

Populations were developed from AC Colibri/OAC Morris (X5154, *n* = 107) and AC Brant/X3145 (X4050, *n* = 102). The seed Cd concentration was analyzed in these RILs along with their parents from plots grown at multiple sites years. The X5154 population was evaluated in Ottawa and Harrow, ON, shown in Figure 1, and the average seed Cd accumulation in Harrow was about an order of magnitude higher than that in Ottawa. The X4050 population was evaluated in Plessisville, QC, and Morden, MB, shown in Figure 2, with higher seed Cd accumulation at Morden compared to Plessisville. The Cd content for some RILs in Harrow and Morden exceeded the upper limit of 0.2 mg kg^−1^ (or 200 parts per billion) for human consumption set by the WHO for food-grade soybean [13,14]. Transgressive segregation for Cd content was observed in both populations under most environments.

### 2.2. Genetic Map and QTL Results

We generated a recombinant map of our F_4_-derived RIL X5154 population, which had a high contrast for Cd, consisting of 325 refined (repeats and monomorphic markers removed) simple sequence repeat (SSR) markers covering all 20 linkage groups. The second recombination map for X4050 was also generated using refined data with 280 SSR markers. Seed Cd concentration and the genetic maps were used for QTL discovery. Our analysis demonstrated that 55% of the variation in X5154 was attributed to a single major QTL between SSR markers SatK139-SaatK150, which corresponds to the Cda1 locus on Gm09 (Lkg-K) reported by two previous studies, Figure 3 [24,25].

When analyzing our second mapping population, X4050 (low × low), we did not find the Cda1 QTL to be responsible for seed Cd concentration. However, a single peak between Satt522 and Satt218 on chromosome Gm13 (Lkg-F) was identified in the 2005 Morden dataset, Figure 4b, which explained 16% of the phenotypic variation. The Plessiville 1999 data, Figure 4a, revealed no QTL but rather a near QTL with a statistical increase of 9% around the same Satt522-Satt218 locus. However, when the means of both sites were used for mapping, the QTL was still visible (14%, Figure 4c). To our knowledge, this minor QTL has not been previously reported and is thought to be a novel QTL corresponding to seed Cd accumulation in population X4050.

### 2.3. Candidate Gene Search

To identify potential candidate gene(s) involved in seed Cd accumulation, a search was conducted between markers Satt522 and Satt218, which are associated with the QTL in population X4050. Using the SoyBase Genome Browser (https://soybase.org/, (accessed on 8 April 2022)), a list of 194 genes were found between the two markers on Chr. 13 (40, 131, 770-41, 821, 873) of the Wm82.a2 genome assembly. The ontologies of these genes were blasted using the SoyBase GO Term Enrichment Tool, and genes with ontologies for Cd response and/or transport and transition ion response and/or transport were identified. Four GO terms that fit our criteria were obtained, along with the identification of seven unique genes (Table 1). Four genes, *Glyma.13G308000*, *Glyma.13G317100*, *Glyma.13G317900*, and *Glyma.13G322100*, were identified as transporters of transition metal ions (GO:0000041). Glyma.13G320800 is involved in the process of metal ion transport (GO:0030001) and divalent metal ion transport (GO:0070838). *Glyma.13G308700* and *Glyma.13G309100* both have biological processes defined as a response to Cd ion (GO:0046686) and are the two primary candidate genes from our results. A review of gene annotations on the soybean physical map in the QTL region identified two transport proteins tightly linked between Satt522 and Satt218. The full list of candidates can be found in Appendix A.

### 2.4. Allele-Specific Marker Development for Cda1

While the previously reported Cda1 locus provides lower seed Cd accumulation when soybean is grown in high Cd soils, pyramiding a second locus, such as the one between Satt522 and Satt218, has the potential to further lower seed Cd accumulation. To facilitate efficient marker-assisted breeding, KASP, CAPS/dCAPS markers were developed for the major QTL Cda1. To accomplish this, two CAPs marker pairs, three dCAPs markers (Appendix A), and one KASP marker (Figure 5) were developed to differentiate the wild-type allele from the mutant in the major Cd gene Cda1 by analyzing 165 lines, including the parental lines for the X5154 and X4050 populations. The results for KASP testing of these soybean lines can be found in Appendix A.

The dCAPS markers supported the findings from the CAPS markers, with the same cultivars showing similar band patterns. From the resulting image for the dCAPS marker (Figure 6b), low seed Cd-accumulating cultivars AC Colibri, Westag, and X3145 were found to have a single band at a relatively small bp when digested with HpyI88I and HinfI, and two bands when digested with TaqI. The high seed Cd-accumulating cultivars AC Hime, AC Proteus, OAC Morris, RD714, and Toki were found to have larger bands at ~242 bp when digested with HpyI88I and HinfI, and only one band at ~147 bp when digested with TaqI.

## 3. Discussion

Seed Cd accumulation is a global issue impacting staple crops, including soybean. Soybean is an essential crop due to its many uses, especially in human food and animal feed; it is thus important for growers to maintain low seed Cd concentrations within this crop. This poses a challenge, as some regions produce soybean seeds in excess of the acceptable Cd levels as defined by WHO [13,14].

In the analysis of seed Cd concentration in the X5154 population (AC Colibri/OAC Morris), a QTL between SatK139 and SaatK150 on Chr. 09 was confirmed (Figure 3). This major QTL for seed Cd content was previously shown to contribute to more than 50% of the parental variation in the populations studied by Jegadeesan et al. and Benitez et al. [24,25], with the underlying gene identified as *Cda1*. With the help of LGC Genomics (LGC group, Teddington, UK), a KASP marker was developed for the major Cd QTL *Cda1* (Figure 5). In addition, CAPS/dCAPS markers were developed (Appendix A, Figure 6) for labs without access to a plate reader as an alternative method for marker-assisted selection. Either *Cda1* marker will be efficient in allowing breeders to create lines with low seed Cd accumulation, as each of the three developed markers reconfirmed data from the other two marker types.

In the analysis of the X4050 population derived from AC Brant/X3145, no QTL was found for *Cda1*. Following subsequent genotyping for *Cda1* (Appendix A and Figure 5), we determined that both parents had the low seed Cd-accumulating allele at *Cda1*, and this locus was not segregating in the X4050 population. We were able to identify a novel minor QTL on Chr. 13, explaining 14% of the parental variation shown in Figure 4c for low seed Cd concentration between SSR markers Satt522 and Satt218. We propose that the newly identified minor Cd QTL might enhance the effectiveness of selecting very low seed Cd-accumulating cultivars when pyramided with *Cda1* [24,25].

GO analysis was conducted to identify candidates among the 194 genes in the Satt522-Satt218 region. Five candidate genes were identified as metal ion transporters using GO analysis. *Glyma.13G308000* encodes an uncharacterized protein of an unnamed family in *G. max* and has no known *Arabidopsis* homolog. It is important to note that, in addition to transition metal ion transport (GO:0000041), this gene has other noteworthy ontologies for this study, including cellular response to iron ion starvation (GO:0010106) and iron ion transport (GO:0006826). *Glyma.13G317100* encodes a Centroradialis-like protein 3 (CEN3) in soybean (*G. max*), a phosphatidylethanolamine-binding protein (PEBP) involved in the negative regulation of flower development by repressing the initiation of floral meristems [26,27,28]. In addition to transition metal ion transport (GO:0000041), the other GOs for *Glyma.13G317100* include negative regulation of flower development (GO:0009910), vegetative to reproductive phase transition of meristem (GO:0010228), and phosphatidylethanolamine binding (GO:0008429). *Glyma.13G317900* is uncharacterized in *G. max*, and the sequence is most closely related to a histidine-containing phosphotransfer protein (AHP) in *Wolffia australiana* and AHP1 in *Arabidopsis*. AHP is part of a two-component phosphorelay involved in MAP kinase cascade regulation [29] and has shown to be expressed in response to plant stresses (i.e., drought, salt, high temperature) [30,31]. In addition to having the ontology of transporter of transition metal ions (GO:0000041), *Glyma.13G317900* also has the biological process GO, an inorganic anion transport (GO:0015698), which lends evidence to suggest this gene may not be the true underlying gene in Cd^2+^ cation uptake. *Glyma.13G322100* encodes a basic helix-loop-helix transcription factor (BHLH56) in soybean (*G. max*). The most closely related *Arabidopsis* homolog is *AtbHLH029* (also known as AtFIT1 Fe-induced transcription factor 1), which is required for the uptake of iron in instances of iron deficiency [32,33]. *Glyma.13G322100* was selected for candidacy for this work on the basis of its physical position on Chr. 13 and its GO term, transition metal ion transport (GO:0000041). However, there is no evidence *Glyma.13G322100* is involved in Cd^2+^ ion transport. Other notable GOs for *Glyma.13G322100* include cellular response to iron ion (GO:0071281) and regulation of iron ion transport (GO:0034756), further indicating the metal ion transported by this gene’s product is iron. *Glyma.13G320800* is identified as a magnesium transporter (mitochondrial RNA splicing protein, MRS2-11) in soybean. MRS2 proteins, in addition to CorA and ALR-type proteins, make up a magnesium (Mg^2+^) transmembrane transporter protein superfamily called MRS2/MGT. This superfamily is the most well-studied Mg^2+^ transporter system and exists widely across plants, animals, bacteria, and fungi, facilitating both the influx and efflux of Mg^2+^ [34,35,36,37,38]. CorA Mg^2+^ transporter proteins are also able to mediate the uptake of cobalt (Co^2+^), manganese (Mn^2+^), and nickel (Ni^2+^) ions [39,40,41]. A study by Li et al. [37] used a ^63^Ni^2+^ tracer inhibition assay to measure the affinity and selectivity of an *Arabidopsis* CorA transport protein, *AtMGT1*, with different divalent cations and found that, while *AtMGT1* had the highest affinity for Mg^2+^, it also had the capacity to transport other divalent cations, including Ni^2+^, Co^2+^, Fe^2+^, Mn^2+^, Cu^2+^, and Cd^2+^ but with a lower affinity than Mg^2+^. While it is likely that the major role of *Glyma.13G320800* is in Mg^2+^ transport, this gene may still provide an avenue for further exploration in response to Cd^2+^ under certain conditions.

Two key candidate genes with GO IDs for response to Cd ion (GO:0046686) were identified: *Glyma.13G308700* and *Glyma.13G309100*. To our knowledge, there are no previously published findings, which indicate either of these two key candidate genes are known to be involved in response to Cd. *Glyma.13G308700* encodes aconitate hydratase (ACO, *acnA*) in soybean (*G. max*), an iron-sulfur protein that functions within the TCA cycle to catalyze the isomerization of citrate to isocitrate. The TAIR10 *Arabidopsis* homolog for *Glyma.13G308700* is aconitase 3 (ACO3). Coleman et al. [42], investigated five different aconitase genes (Aco1, -2, -3, -4, -5) and found *Glyma.13G308700* clustered with the *Arabidopsis* Aco3 gene, *At2g05710,* and suggested that *Glyma.13G308700* may be a paralog of another soybean Aco3 gene, *Glyma.06G305700*, as a result of genetic duplication events. *Glyma.13G308700* has a large number of GOs; in addition to response to Cd ion (GO:0046686), notable GOs include response to iron ion (GO:0010039) and iron ion homeostasis (GO:0055072). Further investigation is needed to determine whether Cd^2+^ ions and Fe^2+^ ions bind the same sites on ACO. *Glyma.13G309100* encodes NADP^+^-dependent 2-alkenal reductase in *G. max.* The most closely related proteins are Phenylpropenal double-bond reductase (PPDBR) from *Pinus pinaster* and alkenal reductase from *Arabidopsis* [42]. Alkenal reductase, an alcohol dehydrogenase-related protein, reduces the carbon–carbon double bonds of α,β unsaturated enones [43] in response to oxidative stress [44]. PPDBRs have been shown to have a role as phenylpropanoid metabolites [45], which are presumed to have a role in plant defense mechanisms, such as biocides and antioxidants [46,47]. In addition to response to Cd ion (GO:0046686), among a sizable list of GOs for *Glyma.13G309100* are toxin catabolic process (GO:0009407) and zinc ion binding (GO:0008270). The mechanism by which the protein encoded by *Glyma.13G309100* catabolizes toxins and the Zn^2+^ ion binding site may be of interest to further investigate Cd uptake.

## 4. Materials and Methods

### 4.1. Population Development and Field Evaluation

Two recombinant inbred line (RIL) populations developed by single seed descent were used in this study. Population X4050 (*n* = 102) (AC Brant [48]/X3145) was originally developed to study seed protein concentration [49]. The parents of X4050 were low for seed Cd content, as shown in Figure 1. In contrast, population X5154 (*n* = 107) (AC Colibri [50]/OAC Morris)) was developed to study Cd content, since the parents contrasted for seed Cd content shown in Figure 2.

Population X4050 was grown in Plessisville, Quebec, in 1999, in a non-replicated modified augmented design as part of the high-protein study [49] and also grown later in Morden, Manitoba, in 2005, in a two-replicate incomplete block design. Population X5154 was grown in 2008 in a non-replicated modified augmented design, and in 2009, in a two-replicate randomized complete block design in Ottawa and Harrow, Ontario.

### 4.2. Sample Preparation and Cd Analysis

Soybean seed Cd analysis followed the procedure outlined by Jegadeesan et al. [24]. Briefly, thirty grams of soybean seed from each plot was washed in deionized water and dried overnight in an oven set to 30 °C prior to being ground for 15 s in a high-speed (20,000 rpm) mill (Knifetec Sample Mill; FOSS, Eden Prairie, MN, USA) equipped with a water-cooled grinding chamber to reduce clumping and ensure a uniform particle size (~0.5 mm). One gram from each sample was digested in a closed-vessel microwave digester with 9 mL ultrapure grade nitric acid and 1 mL of Cd-free H_2_O_2_. After digestion, the sample was added to 50 mL polypropylene tubes and brought to a volume of 50 mL with double-deionized water. Seed Cd concentration in each sample was analyzed by graphite furnace atomic absorbance (SIMMA, PerkinElmer, Waltham, MA, USA), following the recommended procedure of the instrument [51]. Duplicate samples from each plot were performed, and the seed Cd concentration for each line was the mean of the two replicates, with a relative standard deviation of less than 10%.

### 4.3. DNA Extraction, SSR Markers and Primer/SSR and PCR Conditions

Young trifoliate leaves from 5–6 individual seedlings per line from the soybean RIL F_4_ populations X5154 and X4050 were harvested into liquid nitrogen, used in a urea DNA extraction technique described by Molnar et al. [52]; the DNA quality was verified on a UV spectrophotometer, then diluted to a concentration of 20 ng/μL. The SSR markers were designed by Cregan et al. [53] and purchased from Sigma-Aldrich (Sigma-Aldrich Canada Co., Oakville, ON, Canada). A gradient was run to determine the optimum T_m_ of all SSR primers, which ranged from 40 to 66 °C. The SSR primers and T_m_ are listed in Appendix A.

The analysis was performed with 3 μL (20 ng/μL) of genomic DNA, 11.6 μL 5× homemade PCR buffer (400 mM Tris-HCL pH 8.3, 100 mM (NH_4_)_2_SO_4_, 10 mM MgCl_2_, 1 mM DNTPs, 0.5% Triton ×100, H_2_O), 0.12 μL recombinant Taq DNA polymerase (Fermentas; Thermo Fisher Scientific Inc., Wilmington, DE, USA), 0.15 μL SSR primers (20 μM each). The PCR conditions used for amplification of DNA from X4050 and X5154 with SSR primers were as follows: 1 cycle of 3 min at 94 °C; 35 cycles of 45 s at 94 °C, 50 s of T_m_ °C, and 45 s at 72 °C, with 1 final cycle at 72 °C for 5 min upon completion. The PCR products were resolved with 5% polyacrylamide gel electrophoresis, using a 10 bp DNA Ladder CAS 79-06-1 (Invitrogen, Waltham, MA, USA). The gels were visualized using silver nitrate (Fisher, Hampton, N.H.) following Promega’s protocol (www.promega.com, (accessed on 8 April 2022)), and the monomorphic SSR markers were removed from recordings. Only the polymorphic primer pairs were used for the mapping analysis. As a result, a total of 280 and 325 polymorphic SSR markers were selected for populations X4050 and X5154, respectively.

### 4.4. Genetic Map Construction and QTL Analysis

The RIL phenotypes for seed Cd content and SSR markers were incorporated into a genetic map built using Mapmaker Macintosh v2.0.68000 using LOD = 3 and Kosambi’s mapping function to convert recombinant frequencies into map distances (cM). The NQTL computer program (Version 26-Oct-2001, Windows version of MQTL) was applied to the datasets from X5154 and X4050 for QTL detection using a permutation test with 1000 replications at a *p*-value of <0.05. A LOD threshold of 3.0 was set for the trait to declare a QTL. The illustrations of the genetic maps were constructed using MapChart version 2.2 [54].

### 4.5. Candidate Gene Search

A search was conducted for candidate genes for seed Cd accumulation within the newly identified QTL locations through www.soybase.org (accessed on 8 April 2022) utilizing Wm82.a2. The search criteria included any candidate genes that: (1) were located between SSR markers Satt522 and Satt218 and (2) were implicated as a gene that is responsible for Cd or metal transport of any kind. Gene models without known functional families were excluded from the analysis. Genes found within this region were analyzed using the SoyBase GO (gene ontology) Term Enrichment Tool to examine the processes and functions for each gene and to curate a list of candidates. Candidate genes were assessed using the NCBI BLAST (https://blast.ncbi.nlm.nih.gov/Blast.cgi, (accessed on 8 April 2022)) database. The SoyBase Genome Annotation database was used for further annotation of candidate genes, including Top *Arabidopsis* (TAIR10) BLASTP hits [55].

### 4.6. Marker Development for Cda1

A Kompetitive Allele-Specific PCR (KASP) marker was designed for *Cda1* and validated following the KASP manual (LGC Genomics, 2013). Two allele-specific primers, as well as one common reverse primer, were designed by LGC genomics, and the PCR assay was performed using a touch-down protocol of 61–55 °C with 2 re-cycles. Measurements were taken on a Tecan Spark microplate and read using KlusterCaller software (LGC Genomics).

CAPS markers were designed following Primer 3 (http://primer3.ut.ee, (accessed on 8 April 2022)), while the dCAPS markers were designed using dCAPS Finder 2.0 (http://helix.wustl.edu/dcaps/, (accessed on 8 April 2022)). The PCR conditions for both CAPS/dCAPS were: 1 cycle of 3 min at 95 °C; 36 cycles of 45 s at 94 °C, 50 s at T_m_ °C, and 45 s at 72 °C, with 1 final cycle of 5 min at 72 °C upon completion. The primers and restriction enzymes can be found in Appendix A. The PCR was carried out as described previously.

## 5. Conclusions

In the end, we successfully confirmed the presence of a known major gene responsible for low seed Cd accumulation, *Cda1,* in the X5154 population and developed two different markers (CAPS/dCAPS and KASP) for *Cda1* that will facilitate efficient MAS breeding and will allow for selection of very low seed Cd concentration cultivars in the future. We also identified a novel QTL on chromosome 13 in the X4050 population between SSR markers Satt522 and Satt218 with a minor role in seed Cd accumulation, with two potential candidate genes identified. Our findings lay the groundwork for future research on seed Cd accumulation in soybean, enabling further advancement in selecting very low seed Cd-accumulating cultivars that meet international trade limits.

## Figures and Tables

**Figure 1 plants-11-01146-f001:**
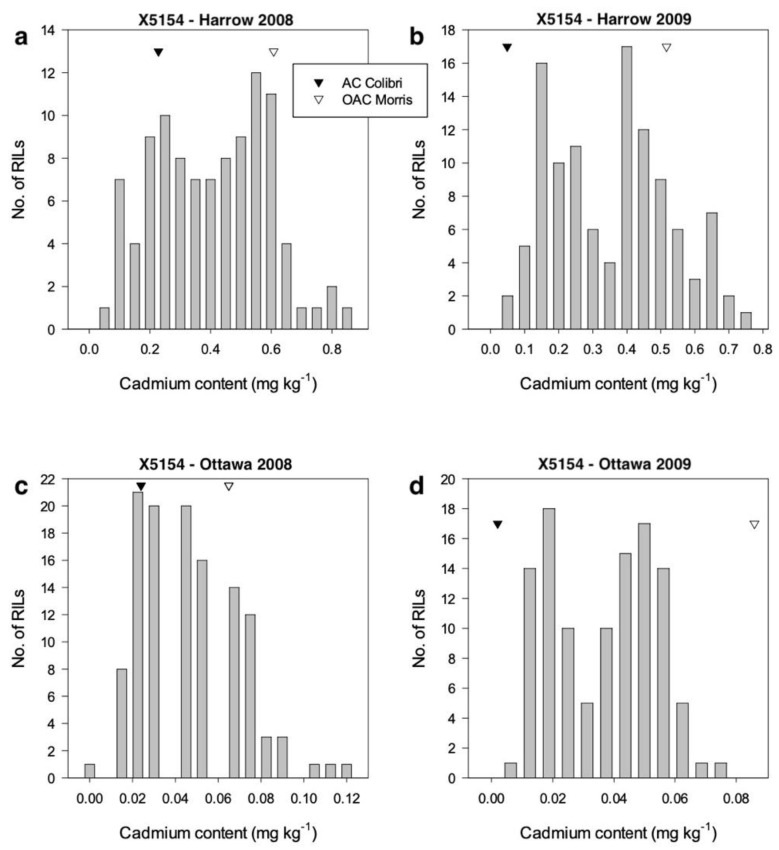
Frequency distribution of seed cadmium accumulation in RIL F_4_ population X5154. (*n* > 100) with parent values from (**a**) Harrow 2008 (se 0.076), (**b**) Harrow 2009 (se 0.063), (**c**) Ottawa 2008 (se 0.019), and (**d**) Ottawa 2009 (se 0.011). se: Standard Error.

**Figure 2 plants-11-01146-f002:**
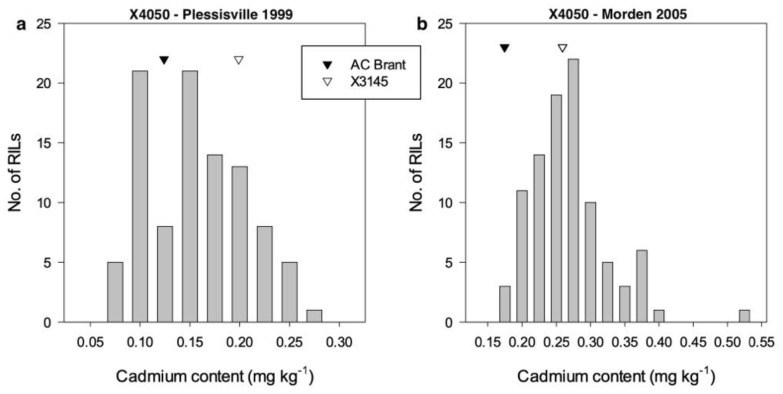
Frequency distribution of seed cadmium accumulation in RIL F_4_ population X4050. (*n* > 100) with parent values from (**a**) Plessisville 1999 (se 0.004) and (**b**) Morden 2005 (se 0.062). se: Standard Error.

**Figure 3 plants-11-01146-f003:**
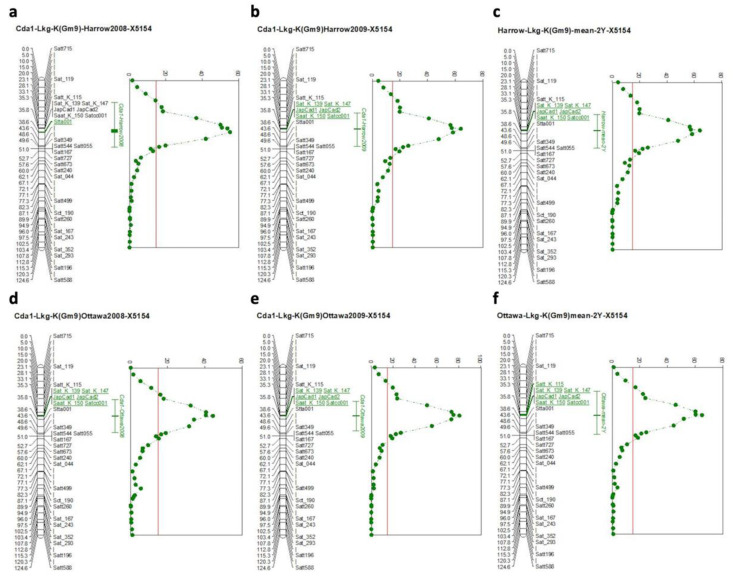
Cd QTL Lkg-K Cda1 for population X5154 (Chr. 09), (**a**) Harrow 2008 with a threshold of 14.6, (**b**) Harrow 2009 with a threshold of 14.8, (**c**) Harrow mean of two years with a threshold of 14.2, (**d**) Ottawa 2008 with a threshold of 14.8, (**e**) Ottawa 2009 with a threshold of 14.9, (**f**) Ottawa mean of two years with a threshold of 14.9. X is the distance in cM, and Y is the test statistic for SIM main effect (test statistic generated by NQTL).

**Figure 4 plants-11-01146-f004:**
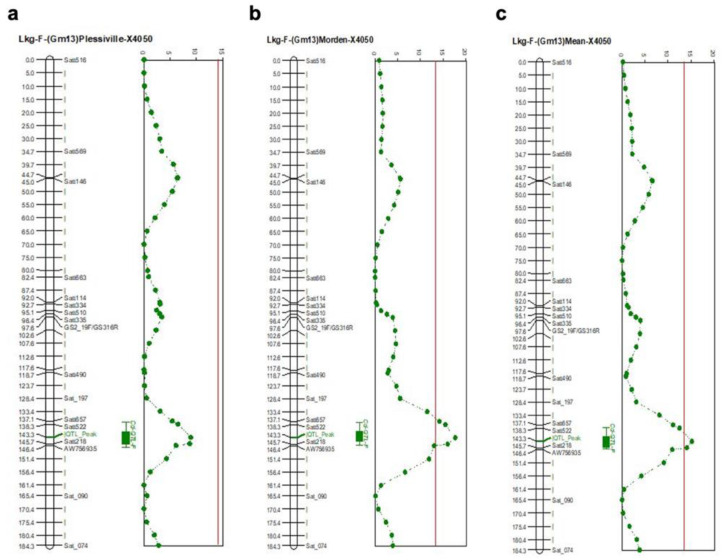
Cd QTL for population X4050 Lkg-F with a threshold of 14.1 for Morden and 13.5 for Plessiville (Chr. 13), (**a**) Plessiville 1999, (**b**) Morden 2005, (**c**) Mean of two sites during two separate years. X is the distance in cM, and Y is the test statistic for SIM main effect (test statistic generated by NQTL).

**Figure 5 plants-11-01146-f005:**
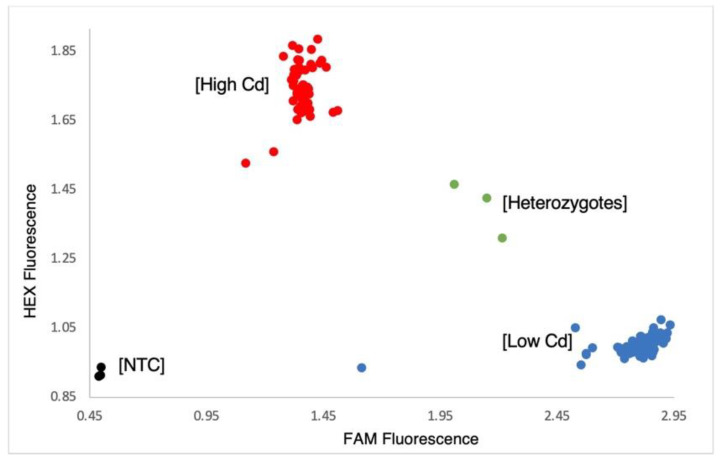
KASP assay screening for Cda1 using HEX and FAM fluorescent fluorophores. Shown in red are the soybean lines with high seed Cd concentration accumulation presented by HEX fluorescence, and in blue are the low seed Cd concentration cultivars presented by FAM fluorescence, while the heterozygotes are shown in green. The no template control (NTC) is shown in black. A total of 165 lines, including parents of the X5154 and X4050 populations investigated by the KASP marker, are presented in Appendix A.

**Figure 6 plants-11-01146-f006:**
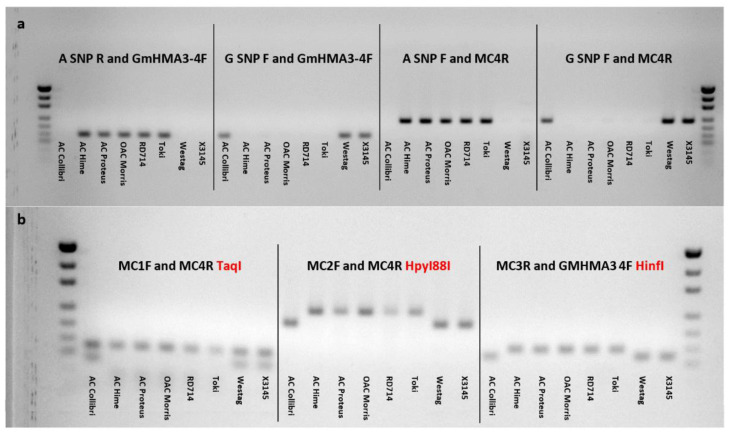
The PCR products of the amplified Cda1 gene among the eight lines. (**a**) High accumulating Cd lines indicated by the bands present among the primer pairs (A SNP R and GmHMA3-4F) and (A SNP F and MC4R), and the low accumulating Cd lines visible by bands present at (G SNP F and GmHMA3-4F) and (G SNP F and MC4R). The amplified digested products for the dCAPS marker pairs; (**b**) low Cd-accumulating lines digested with TaqI resulted in two bands. Lines digested with HpyI88I and HinfI resulted in low Cd accumulators having lower bands compared to the high Cd accumulators. Controls are not shown, using the pUC19 DNA/MspI (HpaII) ladder.

**Table 1 plants-11-01146-t001:** The candidate genes identified within the Satt522-Satt218 region on Chr. 13, along with their corresponding GO terms of interest.

GO_Id	GO_Description	Gene_Id (Wm82.a2.v1)
GO:0000041	Transition metal ion transport	*Glyma.13G308000*
		*Glyma.13G317100*
		*Glyma.13G317900*
		*Glyma.13G322100*
GO:0046686	Response to cadmium ion	*Glyma.13G308700*
		*Glyma.13G309100*
GO:0030001	Metal ion transport	*Glyma.13G320800*
GO:0070838	Divalent metal ion transport	*Glyma.13G320800*

## Data Availability

Not applicable.

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
