# Peer review of "Novel QTL for Low Seed Cadmium Accumulation in Soybean"

_plants, 2022, doi:10.3390/plants11091146_

Round 1
Reviewer 1 Report
Citation: missing (left of the abstract)
line
4 missing * for corresponding author
5-8 missing e-mail addresses
You have more expressions for the same term. Please, choose one expression among them and use it through the entire manuscript.
22 CAPS/dCAPS
24 CAP & dCAP
171,185,186,197,212,368,369,370,377,386,387 CAPS and dCAPS
172 two CAPs marker pairs, three dCAPs markers
34 [3]–[6] > [3–6]
41 [8]–[10] > [8–10]
47 var > var.
54,59,60,62,65,67,68, mg/kg > mg kg-1
54 [13],[14] > [13,14]
60 [16],[17] > [16,17]
62 [17],[18] > [17,18]
65 [19]–[21] > [19–21]
77 [24],[25] > [24,25]
Please, use the same expressions:
87,298 AC Colibri / OAC Morris
207 AC Colibri × OAC Morris
87,296 AC Brant / X3145
217 AC Brant × X3145
87 (X5154, (N=107) > (X5154, N=107)
88 (X4050, N=102)) > (X4050, N=102)
95 [13],[14] > [13,14]
99,103,390 F4 > F4
115,224 [24],[25] > [24,25]
190 ~242bp > ~242 bp
191 ~147bp > ~147 bp
193 different text on the image (Figure 6.a) and below the image (Figure 6.(A)):
image: A SNP R and GmHMA3 4F below: (A SNP R and GmHMA3-4F)
image: A SNP F and HC4R below: (A SNP F and MC4R)
image: G SNP R and GmHMA3 4F below: (G SNP F and GmHMA3-4F)
image: G SNP F and HC4R below: (G SNP F and MC4R)
206 [13],[14] > [13,14]
208 Chr.09 > Chr. 09
210 [24],[25] > [24,25]
221 Chr.13 > Chr. 13
228,239,246,262,273,275,283,359 Arabidopsis > Arabidopsis
234 [26]–[28] > [26–28]
241 [30],[31] > [30,31]
248 [32],[33] > [32,33]
259 [34]–[38] > [34–38]
260 [39]–[41] > [39–41]
287 [46],[47] > [46,47]
392-396 Author Contributions have to be corrected in accordance to the Instructions for Authors.
Author Response
Thank you for your comments on how we can improve the paper.
Citation on the left has been updated.
As for the CAPS and dCAPS comment, we have gone through and changed all the CAPS and dCAPS to CAPS/dCAPS if they are mentioned together. If they are separately mentioned, we left them as is.
The multiple reference formats have also been edited. The mg/kg comment has also been resolved.
Expressions were edited in order to keep only one of the formats.
Finally, figure 6 was updated to include the correct naming in order to match the legend.
Any additional comments suggested by reviewer 1 were also made in the text.
Reviewer 2 Report
In a situation where cadmium contamination is increasing, a segregated group is constructed using varieties with different responses to cadmium content. By using these as population parents to find quantitative trait loci and search for candidate genes in that QTL region, It is considered this article that can be used for breed development and research that can be made in.
However, there are a few things that need to be corrected and there are some questions.
Above all, it is difficult to see the part explained in the figure. It will be easier for readers to see if a title (region, year) is displayed for each graph from a to d in Figure 1, and a to b in Figure 2. The format of Harrow 2008(A) and Harrow 2009(B) or A:Harrow 2008, B:Harrow 2009 is better than (A and B) Harrow 2008 and 2009, respectively.
What SE is, needs an explanation.
The text of the X5154 group presented in Figure 3 is Gm09 (LG K), and the description of the figure is Lkg-K.
Also, Satt499 is quite remote and hard to find.
The X4050 group in Figure 4 is unified as Lkg-F, but as in Figure 3, it would be better to include the title in the figure itself.
As in Figure 3, it would be good to enlarge and display the vicinity of Satt522 ~ Satt218.
I wonder if the QTL of X4050 was not observed in the X5154 group.
It is questionable whether "demonstrated by [22]" in line 71 of the introduction is the correct expression. "demonstrated by Morrison et al. [22]" seems appropriate. The same is true for several subsequent references.
Overall, seed Cd accumulation and Cd accumulation are mixed.
Author Response
Thank you for your comments on how we can improve the paper.
The figure titles at the bottom of Figures 1 and 2 have been updated to be more clear. In addition, figures now have figure titles above the images as well.
In terms of Satt499, we realized this was a mistake on our part as the Cda1 QTL is actually between SatK139-SaatK150 as shown in figure 3 and in reference [24] and not near Satt499. This was fixed in the text and is highlighted in green in figure 3. We would like to thank reviewer 2 for catching this error.
Edits to the text were made based on the remaining comments and suggestions of reviewer 2.
The vicinity of Satt522-Satt218 unfortunately cannot be enlarged due to the method the QTL was made through. If this is an issue, then the whole QTL region would need to be remade in a larger font, but would likely omit some markers which may not be beneficial.
The QTL in X4050 was not observed (it was below the threshold and we decided not to include that) in the X5154 group, which means that pyramiding the new minor QTL with Cda1 could potentially be beneficial for farmers as mentioned in the discussion.